# A prefrontal-bed nucleus of the stria terminalis circuit limits fear to uncertain threat

**Lucas R Glover[1]\*, Kerry M McFadden[1], Max Bjorni[2], Sawyer R Smith[1], Natalie G Rovero[2], Sarvar Oreizi-Esfahani[1], Takayuki Yoshida[1], Abagail F Postle[1], Mio Nonaka[1], Lindsay R Halladay[2], Andrew Holmes[1]**

[1]Laboratory of Behavioral and Genomic Neuroscience, National Institute on Alcohol Abuse and Alcoholism, NIH, Bethesda, United States; [2]Department of Psychology, Santa Clara University, Santa Clara, United States

**Abstract** In many cases of trauma, the same environmental stimuli that become associated with aversive events are experienced on other occasions without adverse consequence. We examined neural circuits underlying partially reinforced fear (PRF), whereby mice received tone-shock pairings on half of conditioning trials. Tone-elicited freezing was lower after PRF conditioning than fully reinforced fear (FRF) conditioning, despite an equivalent number of tone-shock pairings. PRF preferentially activated medial prefrontal cortex (mPFC) and bed nucleus of the stria terminalis (BNST). Chemogenetic inhibition of BNST-projecting mPFC neurons increased PRF, not FRF, freezing. Multiplexing chemogenetics with in vivo neuronal recordings showed elevated infralimbic cortex (IL) neuronal activity during CS onset and freezing cessation; these neural correlates were abolished by chemogenetic mPFC→BNST inhibition. These data suggest that mPFC→BNST neurons limit fear to threats with a history of partial association with an aversive stimulus, with potential implications for understanding the neural basis of trauma-related disorders.

\*For correspondence:
lucasglover@email.gwu.edu

Competing interests: The authors declare that no competing interests exist.

## Introduction

In many cases of psychological trauma, encounters with contexts and stimuli during aversive experience(s) are interleaved with occasions when the same stimuli are experienced without consequence. Most standard rodent assays of fear (i.e., threat) memory, however, present the subject with a conditioning stimulus (CS) that on each occasion is paired with an aversive unconditioned stimulus (US) (*Fanselow and Poulos, 2005*). This discrepancy is pertinent to modeling traumatic memories in rodents, via back translation from human to rodent.

Theoretical accounts of associative learning predict that conditioned responses to CSs with a mixed or partial reinforcement history, which render the CS uncertain or ambiguous with regard to its expected outcome, may differ in certain respects from those that are consistently reinforced. For example, as compared to fully reinforced CSs, partially reinforced CSs can be more difficult to extinguish and produce lesser conditioned responses, due to associative strength accruing to the conditioning context or through the endowment of the CS with inhibitory (CS = no US) properties (*Humphreys, 1939*; *Fitzgerald, 1963*; *Rawlins et al., 1985*; *Rescorla, 2007*; *Tsetsenis et al., 2007*; *Miguez et al., 2012*; *Harris et al., 2019*).

Fear behavior that arises from partial reinforcement could involve neural circuits distinct from the well-described circuits implicated in standard (i.e., fully reinforced) fear conditioning (*Pape and Pare, 2010*; *Bukalo et al., 2014*; *Tovote et al., 2015*). Two brain regions that could be important for the acquisition and expression of partially reinforced fear (PRF) are the medial prefrontal cortex (mPFC, comprising, in the rodent, the prelimbic [PL], infralimbic [IL], and anterior cingulate [ACC]

**eLife digest** While walking home alone late one night, you hear footsteps behind you. Your heart starts to beat faster as you wonder whether someone might be following you. Being able to identify and evade threats is essential for survival. A key part of this process is learning to recognize signals that predict potential danger: the sound of footsteps behind you, for example. But many such cues are unreliable. The person behind you might simply be heading in the same general direction as you. And if you spend too much time and energy responding to such false alarms, you may struggle to complete other essential tasks.

To be useful, responses to cues that signal potential threats must thus be proportionate to the likelihood that danger is actually present. By studying threat detection in mice, Glover et al. have identified a brain circuit that helps ensure that this is the case. Two groups of mice learned to fear a tone that predicted the delivery of a mild footshock. In one group of animals, the tone was followed by a shock on every trial (it was said to be 'fully reinforced'). But in the other group, the tone was followed by a shock on only 50% of trials ('partially reinforced').

After training, both groups of mice froze whenever they heard the tone – freezing being a typical fear response in rodents. But the animals trained with the partially reinforced tone showed less freezing than their counterparts in the fully reinforced group. Moreover, freezing in response to the partially reinforced tone was accompanied by activity in a specific neural pathway connecting the frontal part of the brain to an area called the bed nucleus of the stria terminalis. Inhibiting this pathway made mice respond to the partially reinforced tone as though it had been reinforced on every trial. This suggests that activity in this pathway helps dampen responses to unpredictable threat cues.

In people with anxiety disorders, cues that become associated with unpleasant events can trigger anxiety symptoms, even if the association is unreliable. The findings of Glover et al. suggest that reduced activity of circuits that constrain excessive responses to threats might contribute to anxiety disorders.

cortices) and the bed nucleus of the stria terminalis (BNST) (*Lebow and Chen, 2016*; *Goode et al., 2019*). The mPFC is engaged in experimental situations requiring integration of higher-order cues or disambiguation between conflicting cues to gate a level of response appropriate to the value of outcome (*Sharpe and Killcross, 2018*; *Marek et al., 2019*), while the BNST has been shown to support learning when a stimulus poorly predicts threat (*Lebow and Chen, 2016*; *Goode et al., 2019*).

These structures are also anatomically connected, with a particularly dense connection between the IL and the anterior regions of the BNST (*Hurley et al., 1991*; *McDonald et al., 1999*; *Dong et al., 2001*; *Vertes, 2004*; *Radley and Sawchenko, 2011*; *Radley et al., 2013*; *Johnson et al., 2016*; *Glangetas et al., 2017*; *Tillman et al., 2018*; *Johnson et al., 2019*). Additionally, BNST-projecting IL cells are activated by 'unpredictable' threat in a backward conditioning paradigm (*Goode et al., 2019*). Moreover, stimulation of glutamatergic mPFC inputs produces synaptic depression in the BNST (*Glangetas et al., 2013*). Together, these findings suggest the mPFC and BNST might form a functional circuit regulating fear to ambiguous and uncertain threats.

Here, we sought to elucidate the potential role of the mPFC and BNST and other neural circuits in PRF, using a paradigm in which a CS was paired with a footshock US on only half of the trials (*McHugh et al., 2015*; *Glover et al., 2017*). By combining immediate-early gene mapping, neuronal pathway tracing, in vivo chemogenetics, and a multiplexed approach combining in vivo chemogenetics and in vivo neuronal recordings, we demonstrate that the mPFC→BNST circuit negatively gates PRF.

## Results

### Lower freezing to a partially reinforced CS

The PRF conditioning procedure entailed presenting male C57BL/6J (B6) mice with three pairings of a tone CS and a footshock US, along with three interspersed presentations of the same CS without concomitant footshock (*McHugh et al., 2015*; *Glover et al., 2017*). For comparison, a fully

reinforced fear (FRF) group received 3x CS+US pairings, and a CS-only control group received 6x CS presentations without the US (*Figure 1A,B*).

Freezing increased to a similar extent over the six conditioning trials in the PRF group and over the three conditioning trials, plus the corresponding three no-trial periods, in FRF group, but did not significantly increase in the CS-only group (analysis of variance [ANOVA] group-effect: $F_{(2,17)}=5.74$, p=0.0125; trial-effect: $F_{(5,85)}=13.49$, p<0.0001; interaction: $F_{(5,85)}=1.64$, p=0.1099). On a retrieval test conducted in a novel context (context B) the following day, the PRF and FRF groups froze more than CS-only controls during pre-CS baseline and CS presentation. Notably, however, CS-evoked freezing was lower in the PRF, relative to the FRF, group (ANOVA group-effect: $F_{(2,17)}=53.02$, p<0.0001; CS-effect: $F_{(1,17)}=216.90$, p=0.0001; interaction: $F_{(2,17)}=25.51$, p=0.0001, followed by post-hoc tests: CS-only vs PRF p<0.0001, CS-only vs FRF p<0.0001, PRF vs FRF p=0.0008) (*Figure 1C*, *Figure 1—figure supplement 1*).

These data show that B6 mice express less freezing in the PRF, as compared to FRF, procedure despite the number of CS–US pairings being equivalent in both conditions. These differences are in line with lower freezing in the PRF procedure in a mixed C57BL/6J;CBA/J;129S6/SvEvTac genetic background (*Tsetsenis et al., 2007*) but, indicating a degree of strain dependency of PRF, differ from data in outbred CD-1 mice, in which freezing is equivalent between PRF and FRF groups (*Glover et al., 2017*).

## Mice with an abnormal fear phenotype do not exhibit lower PRF

We next reasoned that an inbred strain (S1), which exhibits impaired contextual (and cued) fear discrimination, deficits in limiting fear following extinction and conditioned inhibition, and high fear expression in a different assay for PRF (*Camp et al., 2012*), might exhibit deficits in the current PRF assay (*Camp et al., 2009*; *Camp et al., 2012*; *Figure 1D*).

Across conditioning trials, there was increased freezing in the PRF and FRF groups, irrespective of strain (ANOVA group-effect: $F_{(1,25)}=0.15$. p=0.7023, trial-effect: $F_{(5,125)}=38.02$, p<0.0001; strain-effect: $F_{(1,25)}=7.68$, p=0.0103; three-way interaction: $F_{(5,125)}=0.66$, p=0.6539). On retrieval in context B, PRF B6 mice showed less CS-related freezing than their FRF counterparts, whereas freezing was equivalent in PRF and FRF S1 mice (ANOVA strain-effect: $F_{(1,27)}=8.72$, p=0.0065; conditioning-type effect: $F_{(1,27)}=6.15$, p=0.0197; CS-effect: $F_{(1,27)} = 524.00$, p<0.0001; three-way interaction: $F_{(1,27)}=7.03$, p=0.0132, followed by post-hoc tests: PRF vs FRF in B6 p=0.0038, PRF vs FRF in S1 p=0.2322) (*Figure 1E*, *Figure 1—figure supplement 1*).

The finding that S1 mice exhibit similar freezing to the PRF and FRF procedures aligns with the excessive fear shown by this strain to innocuous stimuli and following extinction (*Camp et al., 2009*; *Camp et al., 2012*) and further illustrates the strain dependency of PRF.

## Increased latency to feed in the novelty-suppressed feeding test after PRF

An earlier study by *Glover et al., 2017* found that following PRF conditioning, CD-1 mice had a higher latency to feed, as compared to a FRF group, in the novelty-suppressed feeding (NSF) test, an assay sensitive to anxiolytics and antidepressants (*Ramaker and Dulawa, 2017*).

To test whether PRF had a similar effect in B6 mice, NSF was assessed under either high or low illumination levels the day after B6 mice underwent either PRF or FRF. Under high, but not low, illumination, latencies to feed were higher in the PRF and FRF groups than unconditioned controls (ANOVA group-effect: $F_{(1,44)}=10.93$, p=0.0019; illumination-effect: $F_{(1,44)}=4.11$, p=0.0230; interaction: $F_{(1,44)}=2.83$, p=0.0699, followed by post hoc tests: PRF vs Con p=0.0004, FRF vs Con p=0.0222) (*Figure 1—figure supplement 2*).

These data show that PRF conditioning increases anxiodepressive-like anxiety-like behavior under relatively aversive (high illumination) conditions of approach–avoidance conflict.

## Ex vivo neuronal regional activity correlates of PRF

The complex behavioral sequelae of PRF suggest that this form of fear may have different neural substrates than FRF. We, therefore, sought to identify neural correlates of PRF by quantifying the number of c-Fos+ cells, as a proxy for neuronal activity, in forebrain regions following retrieval (for corresponding behavioral data, see *Figure 1B*).

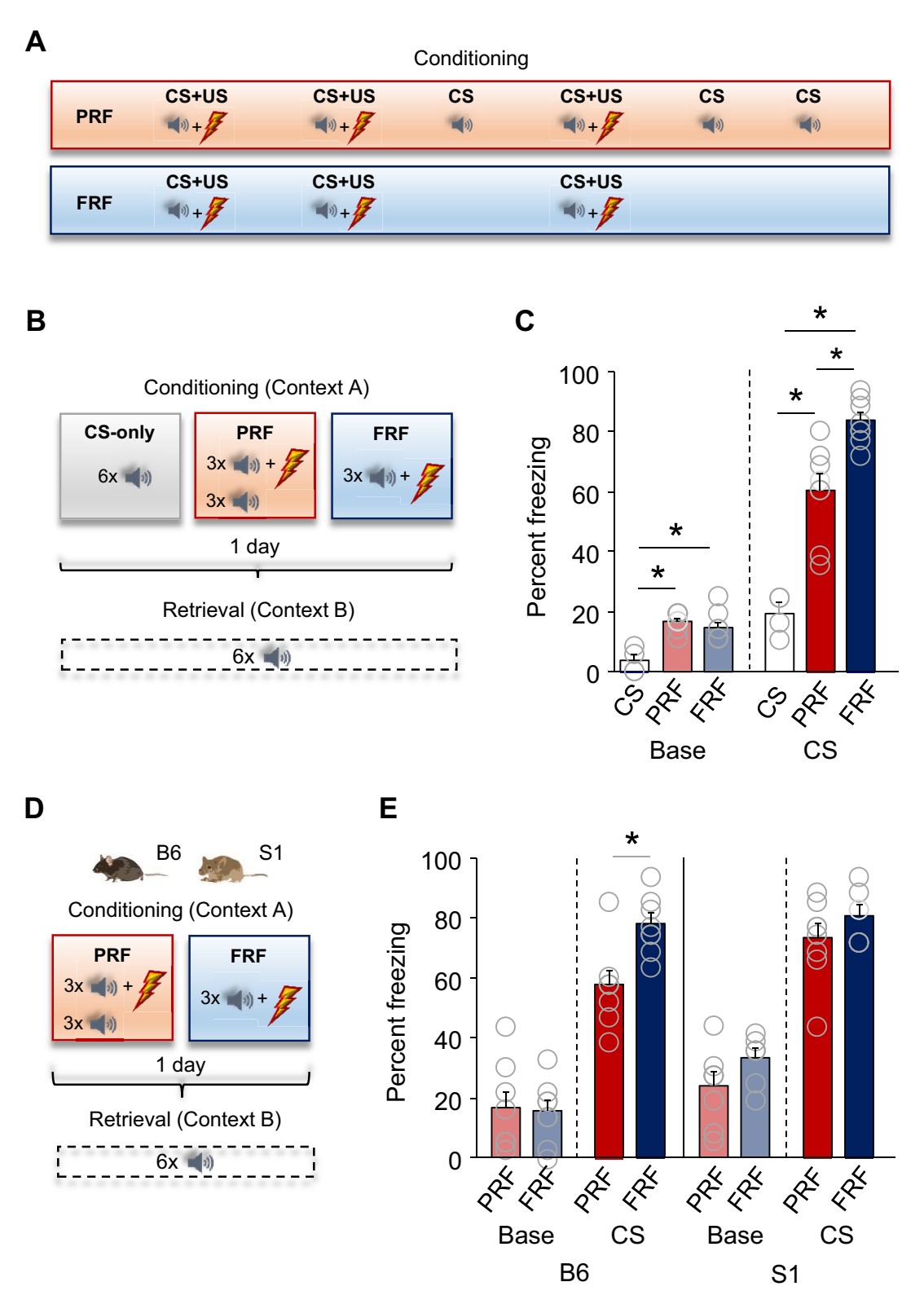

**Figure 1.** Lower freezing during retrieval of partially reinforced fear; effects of genetic strain. (**A**) Schematic depiction of experimental procedure for assessing, in B6 mice, PRF and FRF, along with CS-only controls. (**B**) Schematic depiction of experimental procedure for assessing, in B6 mice, PRF and FRF retrieval in a novel context (context B) and the conditioning context (context A) (**C**) Lower CS-related freezing during retrieval in PRF mice than in FRF mice. Higher baseline and CS-related freezing in PRF and FRF mice relative to CS-only controls (n = 4–8 mice per group). (**D**) Schematic

*Figure 1 continued on next page*

Figure 1 continued

depiction of experimental procedure for assessing PRF and FRF retrieval in the B6 and S1 genetic strains. (E) Lower CS-related freezing during retrieval in PRF than in FRF in B6, not S1, mice (n = 7–8 mice per group/strain). Data are means ± SEM. *p<0.05.

The online version of this article includes the following source data and figure supplement(s) for figure 1:

**Source data 1.** PRF versus FRF (*Figure 1C*).
**Source data 2.** Strain comparison (*Figure 1E*).
**Figure supplement 1.** Freezing during conditioning.
**Figure supplement 2.** Increased latency to feed in the NSF test after PRF.

There were a higher number of c-Fos+ cells in the basolateral amygdala (BLA) of FRF mice (ANOVA group-effect: $F_{(2,17)}=6.79$, p=0.0068, followed by post hoc tests: FRF vs CS-only p=0.0038, FRF vs PRF p=0.0132), as compared to either PRF mice or a set of controls that had received CS-only trials during conditioning. In the paraventricular nucleus of the thalamus (PVT), another region implicated in fear (*Penzo et al., 2015*), c-Fos+ counts were higher in the PRF and FRF groups than controls (ANOVA group-effect: $F_{(2,17)}=4.01$, p=0.0374, followed by post-hoc tests: CS-only vs PRF p=0.0281, CS-only vs FRF p=0.0145). No group differences were evident in the lateral or medial habenula, or ventral or dorsal hippocampus (*Figure 2A–I*, *Figure 2—figure supplement 1*).

In subregions of the mPFC, however, there were more c-Fos+ cells in the IL ($F_{(2,17)}=8.21$, p=0.0032, followed by post-hoc tests: CS-only vs PRF p=0.0009, CS-only vs FRF p=0.0411, FRF vs PRF p=0.0420), but not the posterior ACC ($F_{(2,17)}=1.01$, p=0.3862) of PRF and FRF mice, relative to CS-only controls. Counts in the PL were higher in PRF mice relative to controls and trended higher in the FRF group ($F_{(2,17)}=3.60$, p=0.0499, followed by post hoc tests: CS-only vs PRF p=0.0196). The same pattern of elevated activity in the PRF group, relative to the other groups, was also evident in the BNST, though specifically in the anteroventral BNST (avBNST) ($F_{(2,17)}=19.43$, p=0.0001, followed by post hoc tests: CS-only vs PRF p=0.0001, CS-only vs FRF p=0.0294, PRF vs FRF p=0.0005), not the anterodorsal BNST (adBNST) ($F_{(2,14)}=1.38$, p=0.2831) (*Figure 2A–I*).

These findings show that retrieval of a PRF CS, despite being characterized by lower freezing than FRF, associates with a unique pattern of regional brain activation, with preferentially high activation in the IL and PL subregions of the mPFC and the avBNST.

## Connectivity between mPFC, BNST, and downstream targets

Previous studies in the rat have demonstrated a direct (GABAergic) input from the mPFC to the BNST that is particularly dense between the IL and avBNST (*Dong et al., 2001*), but also present between the PL and avBNST (*Johnson et al., 2016*; *Johnson et al., 2019*). As our c-Fos data indicated activation of the IL, PL, and avBNST by PRF, we sought to verify an mPFC-to-BNST projection in mice.

In a combinatorial viral tracing approach to label postsynaptic targets of mPFC neurons in the BNST (*Zingg et al., 2017*; *Sengupta and Holmes, 2019*), a construct containing a Cre-containing anterograde trans-synaptic virus was infused into the mPFC and a Cre-dependent, synaptophysin-containing, mCherry-fused construct infused into the BNST (*Figure 2—figure supplement 2*). Indicative of monosynaptic input from the mPFC, mCherry labeling was apparent in BNST neurons, mainly in the ventral areas below the anterior commissure. In the rat, PL neurons form close appositions with GABAergic cells in the avBNST that in turn send efferents to the paraventricular nucleus of the hypothalamus (PVN), a key mediator of responses to stress and defensive behaviors (*Johnson et al., 2016*; *Johnson et al., 2019*). Indicating that a similar connection is likely present in mice, inspection of our tissue revealed mCherry/synaptophysin expression originating from mPFC-innervated BNST neurons in the PVN, as well as lateral hypothalamus.

A corollary to the existence of a disynaptic mPFC–BNST–PVN circuit in mice is whether the PVN in turn targets other fear-mediating regions in this species. To gain initial insight into this question, we infused a Cre-dependent, YFP-fused construct containing either channelrohodpsin2 (ChR2) or synaptophysin into the PVN of oxytocin-Cre mice, to label a major population of (oxytocin-positive) PVN cells. This indicated labeling in the ventrolateral periaqueductal gray (vl/PAG) (*Figure 2—figure*

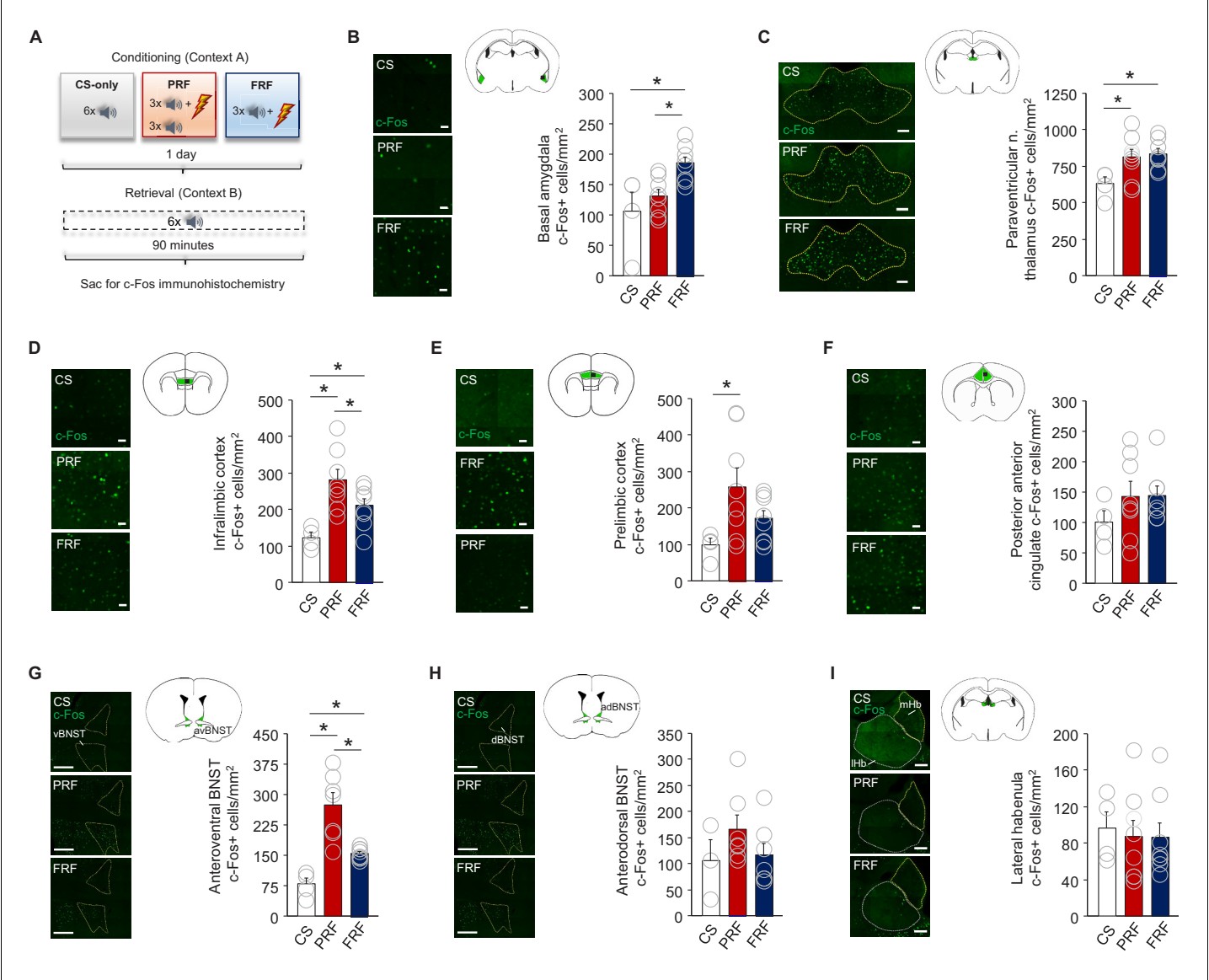

**Figure 2.** PRF preferentially activates subregions of mPFC and BNST. (**A**) Schematic depiction of experimental procedure for assessing ex vivo neuronal regional activity (via c-Fos immunohistochemistry) after PRF or FRF retrieval, along with CS-only controls. Representative images and c-Fos+ cell count differences for basal amygdala (**B**), paraventricular nucleus of the thalamus (**C**), infralimbic cortex (**D**), prelimbic cortex (**E**), posterior portion of the anterior cingulate cortex (**F**), anteroventral BNST (**G**), anterodorsal BNST (**H**), and lateral habenula (**I**). For corresponding behavioral data, see *Figure 1B*. Scale bars = 30 µm (**B,D–F**), 100 µm (**C,I**), 300 µm (**G,H**). n = 4–8 mice per group. Data are means ± SEM. *p<0.05.
The online version of this article includes the following source data and figure supplement(s) for figure 2:

**Source data 1.** c-Fos.
**Figure supplement 1.** Ex vivo neuronal regional activity correlates of PRF.
**Figure supplement 2.** Connectivity between mPFC, BNST, and downstream targets.

supplement 2), a region known to regulate defensive behaviors including freezing (*Tovote et al., 2015*).

Together these data provide evidence of input from the mPFC to the BNST in the mouse, as well as onward connections from the BNST to the PVN and in turn possibly on to the vl/PAG. Thus, PRF engagement of the mPFC and BNST can be viewed in the context of a direct connection between these regions and their downstream access to a broader fear-regulating neural circuitry.

## Inhibition of mPFC→BNST neurons increases freezing to a PRF CS

To causally interrogate the contribution of the mPFC→BNST pathway to PRF, a retrogradely transported Cre-containing construct viral construct was infused into the BNST and a construct containing a Cre-dependent form of hM4Di (or mCherry control) infused into the mPFC, enabling the expression of the inhibitory DREADD in mPFC→BNST neurons to inhibit their activity, via systemic injection of clozapine N-oxide (CNO), during retrieval (*Figure 3A,B*).

During conditioning, freezing increased over trials to a similar extent in all groups (ANOVA trial-effect: F(5,145)=23.54, p<0.0001; group effect: F(3,145)=0.91, p=0.4467; interaction: F(15,145) =0.61, p=0.08647) (*Figure 3—figure supplement 1*). Following CNO administration, CS-related freezing during retrieval was lower in PRF mice than in FRF mice expressing the control virus, replicating our earlier data. By contrast, there was no difference in freezing in mice expressing hM4Di (ANOVA conditioning-type effect: F(1,29)=9.35, p=0.0048; virus-group effect: F(1,29)=12.15, p=0.0016; CS: F(1,29)=1331.02, p<0.0001; three-way interaction: F(1,29)=6.58, p=0.0157, followed by post-hoc tests: mCherry PRF vs mCherry FRF p<0.0001, hM4Di PRF vs hM4Di FRF p=0.1425, mCherry PRF vs hM4Di PRF p=0.0013, mCherry PRF vs hM4Di PRF p=0.7951) (*Figure 3C*). Examination of the trial-by-trial freezing during retrieval indicated no significant trial-related differences in freezing, despite a trend for decreasing freezing across trials in the mCherry PRF group (ANOVA trial-effect: F(5,145)=1.83, p=0.1098; group-effect: F(3,29)=14.15, p<0.0001; trial x group interaction: F(15,145)=1.04, p=0.4213) (*Figure 3—figure supplement 1*).

These data show that inhibition of mPFC→BNST neurons increases freezing to a PRF CS. This finding suggests that engagement of these mPFC→BNST neurons limits the expression ofto the unreliable, PRF, though it remains possible that inhibition of these neurons also produces an increase in PRF expression, which may have been masked due to high (ceiling) levels of freezing.

## IL cells signal CS onset and freezing cessation

The finding that inhibiting mPFC outputs to the BNST pathway increases freezing to a PRF CS implies that mPFC neurons likely encode some aspects of fear. To address this possibility, we devised an approach entailing chemogenetic inhibition of mPFC→BNST neurons (as described above) coupled with in vivo recordings of mPFC single-unit activity via chronically implanted electrode arrays, which we targeted at the IL (*Figure 3—figure supplement 2*). The average firing rate of units did not statistically differ between groups (FRF mCherry: 4.10 ± 0.64, FRF hM4Di: 3.45 ± 0.49, FRF mCherry: 2.67 ± 0.66, FRF hM4Di: 1.67 ± 0.34).

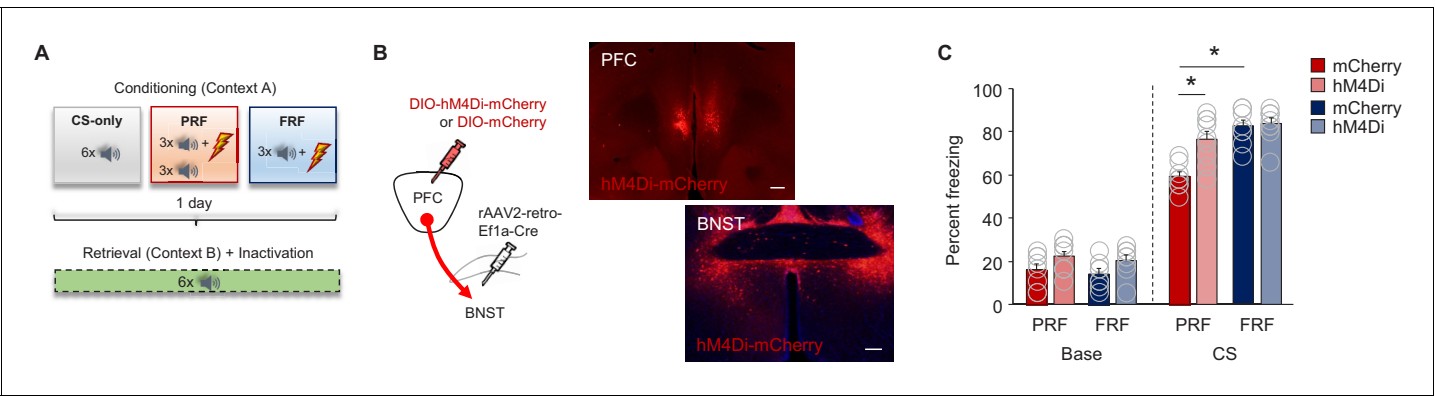

**Figure 3.** Inhibition of mPFC→BNST neurons increases PRF. (**A**) Schematic depiction of experimental procedure for assessing effects of chemogenetic inhibition of mPFC→BNST neurons during retrieval. (**B**) Cartoon of viral strategy and representative images of hM4Di–mCherry labeling in BNST neurons receiving mPFC input (scale bars = 200 μm). (**C**) Lower CS-related freezing during retrieval in PRF mice than in FRF mice transfected with mCherry, not hHM4Di. Data are means ± SEM. *p<0.05.

The online version of this article includes the following figure supplement(s) for figure 3:

**Figure supplement 1.** Freezing during conditioning prior to mPFC→BNST inhibition on retrieval.

**Figure supplement 2.** Electrode placements and virus localization for combined chemogenetic/single-unit recordings.

**Figure supplement 3.** CS and freezing-related IL unit activity and effects of mPFC→BNST inhibition.

**Figure supplement 4.** Heat maps illustrating IL unit activity.

Aligning the single-unit data to the presentation of the CSs during retrieval revealed examples of IL units with activity time-locked to the onset of the CS (*Figure 3—figure supplement 3*). Units exhibiting activity >/<1.96 z scores from baseline in at least two 100 ms time bins within the 500 ms of CS onset were classified as CS responsive (CS-ON). Overall, CS-ON units showed a significant change in neuronal activity in response to the CS (baseline: 0.15 ± 0.35, post-CS: 1.43 ± 0.55, paired t-test: t(10)=6.51, p<0.0001) (*Figure 3—figure supplement 3*, and for heat maps, see *Figure 3—figure supplement 4*). Peak responses occurred within 200–300 s of CS onset and were highest in the mCherry FRF group (*Figure 3—figure supplement 3*). However, when the percentage of CS-ON units was calculated and compared across the conditioning and virus groups, this revealed a higher proportion of CS-ON units in the mCherry groups than in hM4Di groups for PRF mice (Fisher's exact test: p=0.0122), but no differences between virus groups in the FRF mice (Fisher's exact test: p=0.6090), and no difference between PRF and FRF groups, irrespective of virus group (Fisher's exact test in mCherry: p=0.2510; in hM4Di: p=1.000) (*Figure 3—figure supplement 3*).

To examine whether IL cells were also associated with the behavior of mice during testing, their activity was aligned to episodes of freezing and those cells displaying a reliable change relative to either the onset or cessation of freezing (i.e., resumption of movement; >/<1.96 z from baseline in at least two 100 ms time bins within the 500 ms of the event). These units, classified as Freeze-ON and Freeze-OFF, respectively, showed a significant change in baseline-normalized activity (Freeze-ON baseline: −0.81 ± 0.47, post-event: −2.12 ± 0.52, paired t-test: t(10)=4.60, p=0.0010, Freeze-OFF baseline: 0.73 ± 0.43, post-event: 1.67 ± 0.40, paired t-test: t(12)=8.54, p<0.0001) (*Figure 3—figure supplement 3*, and for heat maps, see *Figure 3—figure supplement 4*). Freeze-ON units displayed a decreased firing rate at freezing onset, which was most evident in both of the mCherry groups, while Freeze-OFF units increased firing rate at the cessation of freezing in both groups (*Figure 3—figure supplement 3*).

When the percentage of these cell types were compared across groups, there was a higher percentage of Freeze-OFF units in the mCherry PRF group than in the hM4Di PRF group (Fisher's exact test: p=0.0024), whereas there was no group difference in FRF mice (Fisher's exact test: p=1.000) and no difference between PRF and FRF groups in either the mCherry (Fisher's exact test: p=0.4600) or hM4Di (Fisher's exact test: p=0.0590) virus conditions (*Figure 3—figure supplement 3*). Conversely, there was no difference between the mCherry and hM4Di groups in the proportion of Freeze-ON units, irrespective of whether mice had undergone PRF or FRF.

## Discussion

Here, we sought to provide new insight into the neural substrates regulating the fear response to an uncertain/ambiguous threat. Employing an assay of partial tone+shock reinforcement in B6 mice, we found that PRF conditioning produced a lower fear response than FRF, which was associated with preferential neuronal activation in the mPFC and BNST. We also show that the mPFC and BNST formed a monosynaptic circuit that, when chemogenetically inhibited, caused a selective increase in the expression of PRF and an attendant loss of in vivo correlates of both CS onset and freezing cessation in IL units.

The current findings align with and extend prior work implicating the mPFC and BNST in various situations in which there is ambiguity and uncertainty about a threat. For example, the mPFC is engaged in settings that require integration of higher-order cues to gate learned responses (*Halladay and Blair, 2015*; *Halladay and Blair, 2017*; *Sharpe and Killcross, 2018*; *Marek et al., 2019*), or where there is conflict between excitatory and inhibitory CS associations, for instance in fear extinction (*Milad and Quirk, 2012*; *Bloodgood et al., 2018*; *Lay et al., 2020*), fear discrimination (*Grosso et al., 2018*), threat/safety conditioning (*Sangha et al., 2014*; *Meyer et al., 2019*), and punished reward-seeking (*Burgos-Robles et al., 2017*; *Halladay et al., 2020*).

The BNST, meanwhile, supports learning in the absence of the BLA (*Poulos et al., 2010*; *Zimmerman and Maren, 2011*), and when a stimulus poorly predicts threat (*Lebow and Chen, 2016*; *Goode et al., 2019*; *Bjorni et al., 2020*), either because it is distal (e.g., predator odor; *Fendt et al., 2003*; *Xu et al., 2012*; *Breitfeld et al., 2015*; *Verma et al., 2018*; *Goode et al., 2020*), diffuse (e.g., contextual; *Sullivan et al., 2004*; *Kalin et al., 2005*; *Duvarci et al., 2009*; *Davis et al., 2010*; *Luyten et al., 2012*; *Jennings et al., 2013*; c.f. *Haufler et al., 2013*), or temporally ambiguous (e.g., random or sustained; *Waddell et al., 2006*; *Walker et al., 2009*;

*Hammack et al., 2015*; *Daldrup et al., 2016*; *Goode and Maren, 2017*; *Lange et al., 2017*) with respect to the US.

These known functions of the mPFC and BNST make these structures well placed to mediate fear under conditions of partial reinforcement where the CS is experienced both with and without the US. As we show here, the mPFC and BNST form a discrete neural circuit, through a direct anatomical connection, that serves to limit the expression of partially reinforced fear. This observation is reminiscent of a recent study showing that IL neurons projecting to the avBNST are activated (measured by c-Fos) by a measure of unpredictable threat (backward conditioning), in which US presentation precedes the CS (*Goode and Maren, 2017*). In conjunction with the current data, there is convergent evidence supporting a key role for the mPFC→BNST circuit in mediating fear across various measures of threat uncertainty, ambiguity, and unpredictability.

The precise nature of this role remains to be fully clarified, however. One possibility is that when discrete cues are relatively poor predictors of danger, other environmental stimuli, such as context, modulate the expression of fear in a manner that recruits the mPFC to exert top-down control over the BNST. In support of this possibility, the mPFC is posited to subserve higher-order modulation of conditioned responding (*Sharpe and Killcross, 2018*); indeed, a recent study in rats found that lesions of either the PL or IL impaired one measure of such modulation known as occasion setting (*Roughley and Killcross, 2019*). Another possible explanation for the increase in PRF caused by mPFC→BNST inhibition is that CS-alone presentations during conditioning imbues the CS with inhibitory properties that are gated by mPFC→BNST neurons during retrieval.

Learned inhibition is a function attributed to the mPFC and in particular the IL. For example, pharmacological, optogenetic, and chemogenetic inhibition of the IL impairs the formation and/or retrieval of fear extinction memories (*Laurent and Westbrook, 2009*; *Bukalo et al., 2015*; *Do-Monte et al., 2015*; *Kim et al., 2016*; *Lay et al., 2020*; *Bukalo et al., 2021*) and the expression of learned safety acquired through explicit CS–US unpairing (*Sangha et al., 2014*). Conversely, presentation of a safety signal during inescapable stress decreases activity (c-Fos) in a lateral area of BNST encompassing avBNST (*Christianson et al., 2011*). Furthermore, lesioning this area reduces inappropriate fear to a non-reinforced CS in rats with high-trait anxiety (*Duvarci et al., 2009*). Indeed, a growing number of lesion and functional neuroimaging studies in non-human primates and humans have implicated the BNST in the processing of uncertain threat (see *Goode et al., 2019*; *Miles and Maren, 2019*). Together, these findings suggest that inhibitory properties of the partially reinforced CS could be signaled by the IL downstream to the BNST, thereby limiting the expression of CS-induced fear to a level appropriate to its partial reinforcement history.

It is important to note in this regard that while we targeted our virus infusions to the IL and avBNST – based on prior evidence of a dense anatomical connection between these subregions – the small size and ventral location of these areas meant that viral transfection encompassed parts of the PL and adBNST. The adBNST is engaged by challenges that produce negative affect (*Centanni et al., 2019*), undergoes plastic changes in response to chronic stress (*Conrad et al., 2011*), and, of particular relevance here, is a target of dorsal raphe (*Marcinkiewcz et al., 2016*), BLA (*Lange et al., 2017*), and central amygdala (*Asok et al., 2018*) neurons that sustain fear responses to predictable and unpredictable threats. As such, although the adBNST is a less densely innervated by the mPFC (*Hurley et al., 1991*; *McDonald et al., 1999*; *Dong et al., 2001*; *Vertes, 2004*; *Radley and Sawchenko, 2011*; *Radley et al., 2013*; *Johnson et al., 2016*; *Glangetas et al., 2017*; *Tillman et al., 2018*; *Johnson et al., 2019*), and current as well as prior (*Goode et al., 2019*) c-Fos data do not indicate adBNST activation with uncertain threat, it would be premature to exclude a contribution of this area to PRF.

The possible contribution of BNST-targeting PL neurons to PRF also should not be discounted. Precisely dissociating the roles of PL and IL inputs to BNST in PRF will be an interesting avenue for future work. While the PL has been ascribed a role in promoting FRF via its outputs to the BLA (*Pape and Pare, 2010*; *Dias et al., 2013*; *Bukalo et al., 2014*; *Tovote et al., 2015*), recent work found that optogenetically silencing PL inputs to the avBNST *increased* immobility and associated stress hormone responses in the rat shock-probe burying and tail suspension tests (*Johnson et al., 2019*; see also *Radley et al., 2009*). These effects suggest a role for PL inputs in attenuating negative affect and as such are broadly congruent and potentially explanatory of the current data, despite important differences in methodology.

In the current study, we found neuronal correlates of PRF, specifically within the IL. As in our prior studies of FRF in mice (*Fitzgerald et al., 2014*; *Fitzgerald et al., 2015*), a subset of IL neurons were phasically active to CS presentation during fear retrieval. Intriguingly, we also found a subset of IL neurons that displayed phasic activity during the cessation, but not onset, of freezing during fear retrieval, echoing recordings in rats that uncovered movement-related activity in IL units (*Halladay and Blair, 2015*; *Halladay and Blair, 2017*). Inhibition of BNST-projecting mPFC neurons during fear retrieval essentially abolished the CS- and Freeze-OFF-associated neuronal activity in IL neurons and, in parallel, increased freezing in the PRF group. The ability of mPFC→BNST inhibition to ablate these neuronal correlates could have arisen from the chemogenetic inhibition of BNST-projecting IL neurons, resulting in a loss-of-function in this pathway and a selective increase in PRF mice. Two caveats to this interpretation are that, firstly, inhibition-induced increases in freezing in FRF mice may have been masked by a performance 'ceiling' in the FRF control group and, secondly, hM4Di expression in our recording experiment was not restricted to BNST-projections within the IL, and also encompassed neurons in the PL.

With regard to the broader neural circuitry in which the mPFC→BNST circuit operates to mediate PRF, using trans-synaptic tracing, we found evidence of a disynaptic mPFC→BNST→PVN circuit in mice, as has been reported in rats (*Radley and Sawchenko, 2011*; *Johnson et al., 2016*; *Johnson et al., 2019*). The PVN contains a high density of cells expressing peptide hormones implicated in stress and fear, notably corticotrophin-releasing hormone (CRH) and oxytocin (*Herman and Tasker, 2016*; *Triana-Del Río et al., 2019*). Though CRH-expressing neurons in the PVN receive input from the BNST (*Colmers and Bains, 2018*), it is unclear whether oxytocin-producing cells do so. Interesting nonetheless, we found that mouse PVN oxytocinergic neurons strongly innervate the freezing-regulating vl/PAG, replicating earlier work in cats and naked mole rats (*Holstege, 1987*; *Rosen et al., 2008*). Given the vl/PAG also receives input from a population of avBNST neurons that is, in turn, innervated by the PL (*Johnson et al., 2016*; *Johnson et al., 2019*), these tracing results suggest that in addition to directly innervating the PVN, mPFC→avBNST neurons may also have both direct and indirect (via the PVN) access to the vl/PAG. This positions the circuit to modulate multiple behavioral and neuroendocrine responses to PRF.

In summary, the current study found that B6 mice expressed lower fear to a CS that is partially, rather than fully, reinforced with footshock. Lower PRF expression was not apparent in a mouse strain (S1) deficient in fear discrimination and learned inhibition. Furthermore, c-Fos mapping revealed PRF preferentially recruited the mPFC and BNST, and neuronal tracing showed direct neuronal projections from mPFC to the BNST, with downstream connections to stress- and fear-mediating regions. Demonstrating the causal importance of the mPFC→BNST neurons, inhibiting this pathway increased PRF and abolished neuronal correlates of CS presentation and freezing cessation in the IL. Collectively, these findings provide novel insight into the neural substrates of PRF, with potential translational relevance to anxiety and trauma- and stressor-related disorders in which threats are typically ambiguous and unpredictable.

## Materials and methods

### Subjects

Subjects were adult male C57BL/6J (B6), 129S1/SvImJ (S1), and B6;129S-*Oxt*^tm1.1(cre)Dolsn^/J (JAX strain 024234) (Oxt-Cre) mice obtained from the Jackson Laboratory (Bar Harbor, ME, USA) and were at least 8 weeks old at the time of testing. Mice were group-housed in a temperature ($22 \pm 3°$C) and humidity ($45 \pm 15\%$) controlled vivarium under a 12 hr light/dark cycle (lights on 0600 hr). Mice undergoing surgery for chronic implantation were single housed after surgery to prevent the implant being damaged by a cage mate. All experimental procedures were approved by the National Institute on Alcohol Abuse and Alcoholism (NIAAA) and Santa Clara University Animal Care and Use Committees (SCU AWA: D18-01042) and followed the NIH guidelines outlined in 'Using Animals in Intramural Research' and the local Animal Care and Use Committees.

### Partially versus fully reinforced threat (standard procedure)

The threat conditioning procedures were based on previous studies with slight modifications (*McHugh et al., 2015*; *Glover et al., 2017*). For this and all other experiments, prior to testing, mice

were randomly assigned to experimental groups and habituated to handling for approximately 10 min per day for 4 days. The following procedures were used in all experiments, unless stated otherwise below.

Conditioning was conducted in a 27 × 27 × 11 cm chamber with opaque metallic walls and a metal rod floor (context A). The walls of the chamber were cleaned with a 79% water/20% ethanol/1% vanilla-extract solution to provide a distinctive odor – this was repeated after each session. All conditioning procedures began with a 180 s baseline period. Conditioning for FRF entailed three presentations (60–90 s variable inter-CS interval) of a 30 s, 75 dB (50 ms rise time), white noise (CS) that co-terminated with a 2 s, 0.6 mA scrambled footshock (US). After the final pairing for all groups, there was a 120 s no-stimulus period before the mouse was returned to the home-cage. The procedure was the same for PRF, with the exception that the CS was presented, without the US, on an additional three occasions during the intervals between the CS+US pairings (order: CS+US; CS+US; CS-noUS; CS-noUS; CS+US; CS-noUS, 15–60 s inter-stimulus interval). Where a CS-only control group was included (stated below), the CS was presented on six occasions, corresponding to the order and timing of the PRF group, but without any concomitant US.

CS retrieval took place one day after conditioning in a novel context B, a 27 × 27 × 11 cm chamber with white Plexiglas walls (rear wall curved) and a solid white floor, which was housed in a different room to context A. Between each session, all surfaces of the chamber were cleaned with a 99% water/1% acetic acid solution. After a 180 s baseline period, there were six CS presentations (20–60 s inter-pairing interval). There was a 20 s no-stimulus period before the mouse was returned to the home-cage. All groups were tested in the same manner.

Stimulus presentation was controlled by the Med Associates VideoFreeze system (Med Associates, Burlington, VT, USA). Freezing, scored manually every 5 s (as no visible movement except that required for breathing), was measured as an index of fear (*Blanchard and Blanchard, 1972*) and converted to a percentage ([number of freezing observations/total number of observations] x 100).

## Novelty-suppressed feeding

Mice (assigned to FRF, PRF, and a control group exposed to the conditioning context for 1 min) underwent conditioning and were then food-deprived for 24 hr. Subsequently, they were assessed using the NSF test for anxiodepressive-like behavior, as previously described (*Glover et al., 2017*). The test apparatus was a novel, 50 cm$^3$ white Plexiglas box with the floor covered by fresh cage substrate. A single pellet of regular home-cage food chow was placed within a plastic weigh-boat in the center of the box. Separate groups of mice underwent the test under 180 lux (low) and 1350 lux (high) illumination. The mouse was placed in a corner of the box, facing the center, and the latency to begin eating the chow was measured from a video-recording. The test ended when eating started or when 600 s had elapsed.

## Behavior in mouse strain with a persistent and generalized fear phenotype

B6 and S1 mice underwent conditioning and retrieval as described under 'Partially versus fully reinforced threat (standard procedure)'.

## Regional patterns of fear-related c-Fos activity

Mice (assigned into FRF, PRF, and a CS-only group) underwent conditioning and retrieval as described under 'Partially versus fully reinforced threat (standard procedure)'. Ninety minutes after retrieval, mice were deeply anesthetized with sodium pentobarbital and transcardially perfused with ice-cold phosphate-buffered saline (PBS, pH 7.4) followed by ice-cold 4% paraformaldehyde (PFA). Brains were removed, and 50 μm coronal sections were cut on a vibratome (Leica VT1000 S, Leica Biosystems Inc, Buffalo Grove, IL, USA) and stored free floating in 0.1 M phosphate buffer (PB) at 4°C for <1 week.

Sections were incubated successively with 10% normal goat serum and 1% bovine serum albumin in PBS-TritonX (0.3%) for 2 hr, a mixture of rabbit anti-c-Fos (9F6) (cat# 2250S, 1:1000, Cell Signaling Technology, Danvers, MA, USA) and a mouse monoclonal anti-NeuN antibody (MAB377, Millipore, 1:1000) in a dilution of 1% normal goat serum and 0.1% bovine serum albumin in PBS-TritonX (0.3%) for two nights on a platform rocker at 4°C. Sections were then rinsed 3× for 10 min in PBS and

incubated in anti-rabbit Alexa 488 secondary antibody (cat# A-11034, 1:500, Invitrogen, Eugene, OR, USA) and Alexa Fluor 555 anti-mouse antibody (cat# A-21422, 1:500, Invitrogen) in a dilution of 1% normal goat serum and 0.1% bovine serum albumin in PBS-TritonX (0.3%) at room temperature on a platform rocker for 2 hr. Sections were rinsed in PBS 2× for 10 min and then counterstained with Hoechst 33342 (5 µg/mL, cat# H1399, Thermo Fisher Scientific, Waltham, MA, USA) in PBS. Sections were rinsed 3× for 10 min in PBS before each series. After rinsed once in 0.1 M PB for 10 min, serial sections were mounted onto slides, air-dried, coverslipped with aqueous mounting media (10 mM Tris–HCl [pH 8.0] [5 mL], DABCO [cat# D27802-25G, Sigma–Aldrich] [1.42 g], and glycerol [cat# 5516, Sigma–Aldrich] [50 mL]), then sealed with clear nail polish.

Images of all three channels (c-Fos, NeuN, Hoechst) for all sections were acquired using an Olympus VS120 Virtual Slide Microscope system (Olympus, Center Valley, PA, USA, VS_ASW software) with a 20× objective (U Plan S Apo; 20×, NA 0.75). The NeuN channel, in the autofocus mode, was used as a focus reference, in the autofocus mode. For image analysis, the FIJI (https://imagej.net/Fiji) (*Schindelin et al., 2012*) with VSI reader plugin (BIOP, Zurich, Switzerland, https://c4science.ch/w/bioimaging_and_optics_platform_biop/image-processing/imagej_tools/ijab-biop_vsireader/) was used. A contour of each brain area (region of interest, ROI) was manually drawn on the Hoechst channel with reference to a mouse brain atlas (*Paxinos and Franklin, 2001*) on the thumbnail image that covers the whole coronal sections and the full resolution image of the ROI was extracted for all channels.

Counts were made in the following brain regions: PL, IL, ventromedial BNST, dorsolateral BNST, BLA, lateral and medial habenula, and ventral and dorsal hippocampus (for cartoons depicting region definitions and example images, see *Figure 2*, *Figure 2—figure supplement 1*). For each brain region, cell counts were conducted (blind to test group) in two to four sections from each hemisphere, for a total of six data points per region per mouse. It was unnecessary to correct for double counting because sections were non-consecutive. The ROIs were transferred to the c-Fos channel, and the mean number of c-Fos positive cells per $0.25\ mm^2$ within the ROI was quantified in a semi-automated manner using a custom-written macro.

## Trans-synaptic tracing of mPFC→BNST neuronal outputs to hypothalamus

Mice were placed in a stereotaxic alignment system (Kopf Instruments, Tujunga, CA, USA) and kept under isoflurane anesthesia. AAV1-hSyn Cre-WPRE-hGH (titer: $3.5 \times 10^{13}$ GC/mL, plasmid# 55637, obtained from Addgene, Cambridge, MA, USA, and packaged by Vigene Biosciences, Rockville, MD, USA) was unliterally infused (0.15 µL) into the PFC, and AAV5-Ef1a-DIO-eYFP (titer: $2.1 \times 10^{12}$, obtained from the UNC Vector Core) was unilaterally infused (0.15 µL) into the BNST of the same hemisphere. The coordinates for BNST infusions were Medial/Lateral = ±0.80, Dorsal/Ventral = −4.15, Anterior/Posterior = +0.03. The coordinates for mPFC infusions were Medial/Lateral = ±1.30 (20° angle), Dorsal/Ventral = −3.00, Anterior/Posterior = +2.00.

Four weeks later, mice were terminally anesthetized with sodium pentobarbital (50–60 mg/kg). Brains were removed and initially suspended in 4% PFA overnight and then at 4°C in 0.1 M PB for 1–2 days. The general histological procedures were also the same as described under 'Regional patterns of fear-related c-Fos activity', with the exception that sections were successively immunostained with rabbit anti-DsRed (1:200 dilution, cat# 632496, Takara Bio, Mountain View, CA, USA) and Alexa Fluor 555 Goat Anti-Rabbit (1:500 dilution, cat# A-21428, Thermo Fisher Scientific), PBS (9 mL), 10% Triton X-100 (0.3% final) (300 µL), and blocking buffer (as above; 1 mL), and incubated on a platform rocker for 2 hr (20°C). The sections were mounted onto slides and allowed to dry.

Far-Red nuclear staining dye (four to five drops of NucRed Dead 647 ReadyProbe Reagent, Thermo Fisher Scientific, and 0.1 M phosphate buffer [2.5 mL]) was pipetted onto the mounted sections. After 15–20 min, the excess solution was suctioned using a benchtop aspirator. Once sufficiently dried, the slides were coverslipped using the same aqueous mounting media. Fluorescent images were taken with a Zeiss (LSM 700, Carl Zeiss Microscopy, Thornwood, NY, USA) confocal microscope under a Plan-Apochromat 10x/0.8 M27 objective.

## Output tracing of oxytocin PVN cells

Oxt-Cre mice were placed in a stereotaxic alignment system (Kopf Instruments) under isoflurane anesthesia. Either a viral vector-containing ChR2, fused to GFP (AAV2-EF1a-DIO-hChR2(E123T/T159C)-EYFP, titer: $6.10 \times 10^{12}$ vp/mL, Addgene plasmid#35509, obtained from the UNC Vector Core, Chapel Hill, NC, USA) or a vector-containing synaptophysin, fused to GFP (AAV8.2-hEF1a-DIO-synaptophysin-EYFP, titer: $2.1 \times 10^{13}$ vg/mL, generously provided by Dr. R. Neve, Massachusetts General Hospital, Belmont, MA, USA) was bilaterally infused (0.15 µL) into the PVN (coordinates ML = ±1.50 mm (15° angle), DV = −4.87 mm, and AP = +0.75 mm, relative to bregma).

Five weeks later, mice were deeply anesthetized with sodium pentobarbital and transcardially perfused with PBS followed by 4% PFA. Coronal (50 µm thick) sections were prepared by vibratome (VT1000S; Leica). The general histological procedures were also the same as described under 'Regional patterns of fear-related c-Fos activity', with the exception that sections were successively immunostained with chicken anti-GFP (1:5000 dilution, cat# ab13970, Abcam, Cambridge, UK) and anti-chicken Alexa 488 secondary antibody (1:500 dilution, cat# ab150169, Abcam). Images were taken with a fluorescence microscopy (VS120; U Plan S Apo; 20×, NA 0.75; Olympus).

## Effects of in vivo chemogenetic mPFC→BNST pathway inhibition

Mice were placed in a stereotaxic alignment system (Kopf Instruments) and kept under isoflurane anesthesia. rAAV2-retro-Ef1a-Cre (titer: $1.0 \times 10^{13}$ gc/mL, obtained from the Salk Institute, La Jolla, CA, USA) was bilaterally infused targeting the BNST (0.15 µL/hemisphere). Additionally, either AAV8-hSyn-DIO-hM4D(Gi)-mCherry-WPRE (titer: $2.25 \times 10^{13}$ gc/mL, obtained from the Massachusetts General Hospital Gene Delivery Technology Core, Cambridge, MA, USA) or AAV8.2-hEF1-DIO-mCherry-WPRE (titer: $2.13 \times 10^{13}$ vg/mL, obtained from the Massachusetts General Hospital Gene Delivery Technology Core) was bilaterally infused targeting the IL (0.15 µL/hemisphere). Each infusion was done over 10 min using a Hamilton syringe and 33-gauge needle. The needle was left in place for a further 5 min to ensure diffusion. The coordinates for mPFC and BNST infusions were as described above.

Four weeks after surgery, mice underwent conditioning and retrieval testing as described under 'Partially versus fully reinforced threat (standard procedure)'. Thirty minutes prior to the retrieval test, CNO (0.01 mg/mL/kg) was injected intraperitoneally.

After the completion of testing, mice were terminally anesthetized with sodium pentobarbital (50–60 mg/kg). Brains were removed and suspended in 4% PFA overnight and then at 4°C in 0.1 M PB for 1–2 days. Coronal sections (50 µm thick) were cut with a vibratome (Leica VT1000 S, Leica Biosystems Inc) and coverslipped with Vectashield HardSet mounting medium with DAPI (Vector Laboratories, Inc, Burlingame, CA, USA). Sections were imaged using an Olympus BX41 microscope (Olympus America Inc, Center Valley, PA, USA). Mice without viral (i.e., mCherry) expression in the region of interest were removed from the analysis.

## In vivo mPFC single-unit recordings during chemogenetic mPFC→BNST inhibition

Mice were placed in a stereotaxic alignment system (Kopf Instruments) and kept under isoflurane anesthesia. rAAV2-retro-Ef1a-Cre (titer: $1.0 \times 10^{13}$ GC/mL, obtained from the Salk Institute) was bilaterally infused targeting the BNST (0.25 µL/hemisphere). In addition, either AAV8-hSyn-DIO-hM4D(Gi)-mCherry-WPRE (titer: $2.25 \times 10^{13}$ GC/mL, obtained from the Massachusetts General Hospital Gene Delivery Technology Core) or AAV8.2-hEF1-DIO-mCherry-WPRE (titer: $2.13 \times 10^{13}$ vg/mL, obtained from the Massachusetts General Hospital Gene Delivery Technology Core) was bilaterally infused targeting the IL (0.25 µL/hemisphere). Infusions were done over 10 min using a Hamilton syringe and 33-gauge needle. The needle was left in place for a further 5 min to ensure diffusion. The coordinates for mPFC and BNST infusions were as described above. During the same surgery, a microelectrode array (two rows of eight electrodes with 35 µm electrode spacing and 200 µm row spacing [Innovative Neurophysiology, Durham, NC, USA]) was unilaterally (hemisphere counterbalanced) targeting the IL (array center: ML = ±0.30 mm, DV = −2.70 mm, AP = +1.75 mm) and affixed to the skull with dental cement.

Five weeks after surgery, mice were habituated to the recording tethers for 30 min in their home-cage for two consecutive days prior to behavioral testing. Mice underwent conditioning and retrieval

testing using the standard procedure described above, with the exception that retrieval was conducted in a 30 cm diameter clear acrylic cylinder with an open top to accommodate the cable connecting the head-stage. Thirty minutes prior to the retrieval test, CNO (0.01 mg/mL/kg) was intraperitoneally injected. Electrophysiological and behavioral recordings were acquired using SpikeGadgets main control unit and Trodes software (SpikeGadgets, San Francisco, CA, USA). Unit recordings were carried out using 16-channel digitizing head-stages, sampled at 30 kHz. Behavioral videos were scored offline by an experimenter blind to conditions.

Single units were sorted manually using Offline Sorter v3.0 (Plexon Inc, Dallas, TX, USA) and analyzed using NeuroExplorer, version 5 (Nex Technologies, Colorado Springs, CO, USA) as previously described (*Halladay and Blair, 2015*; *Halladay et al., 2020*). Unit data were aligned to CS and freezing events. Freezing was manually scored (by an experimenter blind to experimental group) and resultant time stamps aligned with the neuronal data. Freezing onset was defined as a transition from movement to no visible movement except that required for breathing. Freezing cessation was defined as a transition from freezing to movement. To determine whether units were responsive to the CS, data during a 500 ms window following the start of the CS for each unit were binned in 100 ms bins and normalized to a 1 s baseline defined as the 10 bins immediately prior to the start of the CS. Units with at least two bins of the same sign in the 500 ms following the start of the CS with a value of >1.96 ($p<0.05$) were considered significantly different from baseline and classified as CS responsive. Unit responsiveness to freezing onset and freezing cessation were analyzed similarly, with the exception that the baseline was shifted from −2 to −1 (rather than −1 to 0) s prior to start of an onset or cessation event to ensure that event and the baseline were temporally separate.

On completion of testing, mice were anesthetized with 2% isoflurane and a current stimulator (S48 Square Pulse Stimulator, Grass Technologies, West Warwick, RI, USA) that delivered 2 s of 40 µA DC current through each electrode to make a small marking lesion. The next day, mice were overdosed via an intraperitoneal injection of 150 mg/kg Euthasol (Henry Schein, Melville, NY, USA) and perfused intracardially with PBS followed by 4% PFA. Brains were left in 4% PFA overnight, then transferred to a 30% sucrose PBS solution for cryoprotection. Coronal sections (50 µm thick) were cut on a cryostat (Leica Biosystems Inc) and mounted onto slides. Tissue was stained with DAPI (Sigma–Aldrich) and imaged using a Keyence BZ-X800 fluorescence microscope (Keyence Corporation of America, Itasca, IL, USA). Mice without viral (i.e., mCherry) expression in the mPFC or correct electrode placement in the IL were removed from the analysis.

## Statistical analysis

Differences in freezing and c-Fos counts were analyzed using ANOVA followed by Dunn's post hoc tests. Differences in z scored single-unit values were analyzed using paired t-tests. Differences in the percentage of recorded units responsive to the CS onset, freezing onset, and freezing cessation were analyzed using non-parametric Fisher's exact tests. The threshold for statistical significance was set at $p<0.05$; significance values are shown up to $p<0.0001$.

## Acknowledgements

We are very grateful to Dr. L Ostroff, Dr. N Justice, Dr. S Maren, and Dr. TL Kash for valuable discussions. Research supported by the NIAAA Intramural Research Program.

## Additional information

### Funding

| Funder | Grant reference number | Author |
| --- | --- | --- |
| National Institute on Alcohol Abuse and Alcoholism | NIAAA-IRP | Andrew Holmes |

The funders had no role in study design, data collection and interpretation, or the decision to submit the work for publication.

## Author contributions
Lucas R Glover, Takayuki Yoshida, Abagail F Postle, Mio Nonaka, Formal analysis, Investigation, Methodology, Writing - review and editing; Kerry M McFadden, Max Bjorni, Natalie G Rovero, Investigation; Sawyer R Smith, Sarvar Oreizi-Esfahani, Investigation, Writing - review and editing; Lindsay R Halladay, Formal analysis, Supervision, Investigation, Methodology, Writing - original draft, Writing - review and editing; Andrew Holmes, Conceptualization, Data curation, Formal analysis, Funding acquisition, Investigation, Methodology

## Author ORCIDs
Lucas R Glover [iD] https://orcid.org/0000-0002-1127-3819
Takayuki Yoshida [iD] https://orcid.org/0000-0002-2842-2569
Mio Nonaka [iD] https://orcid.org/0000-0002-3492-6659
Lindsay R Halladay [iD] https://orcid.org/0000-0003-2232-6709
Andrew Holmes [iD] https://orcid.org/0000-0001-7308-1129

## Ethics
Animal experimentation: All experimental procedures were approved by the NIAAA (protocol # LBGN-AH-01) and Santa Clara University (SCU AWA: D18-01042) Animal Care and Use Committees and followed the NIH guidelines outlined in 'Using Animals in Intramural Research' and the local Animal Care and Use Committees.

## Decision letter and Author response
Decision letter https://doi.org/10.7554/eLife.60812.sa1
Author response https://doi.org/10.7554/eLife.60812.sa2

# Additional files

## Supplementary files
• Transparent reporting form

## Data availability
Some of the data generated or analysed during this study are included in the manuscript and supporting files. Source data files have been provided for Figure 1. Further data has been uploaded to Dryad (https://doi.org/10.5061/dryad.j9kd51cbn).

The following dataset was generated:

| Author(s) | Year | Dataset title | Dataset URL | Database and Identifier |
|---|---|---|---|---|
| Glover LR, McFadden KM, Bjorni M, Smith SR, Rovero NG, Oreizi-Esfahani S, Yoshida T, Postle AF, Nonaka M, Halladay LR, Holmes A | 2021 | mCherry and Gi DREADD electrophysiology for freezing levels during retrieval | https://doi.org/10.5061/dryad.j9kd51cbn | Dryad Digital Repository, 10.5061/dryad.j9kd51cbn |

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
