## [Decision Letter]

**Acceptance summary:**

The paper by Glover and colleagues elegantly combines behavioral, immunohistochemistry, chemogenetic and recording techniques to elucidate the neural mechanisms of partial compared to continuous reinforcement. The neural regulation of partial reinforcement in the fear domain studied in this paper links well with extensive work done in the field of extinction of fear. Therefore, the discoveries reported in this paper will be sure to guide future research directions in the field.

**Decision letter after peer review:**

Thank you for submitting your article "A prefrontal-bed nucleus of the stria terminalis circuit limits fear to uncertain threat" for consideration by *eLife*. Your article has been reviewed by three peer reviewers, including Mihaela D Iordanova as the Reviewing Editor and Reviewer #1, and the evaluation has been overseen by Kate Wassum as the Senior Editor. The following individuals involved in review of your submission have agreed to reveal their identity: Michael A McDannald (Reviewer #2); Rebecca Shansky (Reviewer #3).

The reviewers have discussed the reviews with one another and the Reviewing Editor has drafted this decision to help you prepare a revised submission.

Summary:

In a series of studies, Glover and colleagues examined the role of the prefrontal cortex and its input to the bed nucleus of stria terminalis pathway in fear to uncertain threats. The authors first demonstrate that partially reinforced cues support lower fear levels than fully reinforced cues, this effect is context-specific and is dependent on mouse strain. Comprehensive c-fos mapping is used to show that differential activation of the prefrontal cortex (prelimbic [PL] and infralimbic regions [IL]) and the bed nucleus of the stria terminalis (BNST) in threat uncertainty. DREADD inhibition of BNST-projecting mPFC neurons increased freezing to the partially reinforced cue. Electrophysiological examination revealed that IL neurons are preferentially activated in the PRF group at CS-ON and the number of these neurons is greatly reduced when the mPFC-BNST pathway is inactive. All reviewers agreed that the data presented in the manuscript are exciting and of great importance in that it uncovered a novel role for the mPFC-BNST pathway in threat uncertainty. The multifaceted approach is also a major strength.

Through the review process a number of essential revisions were identified. These are specified below. None of the revision require additional data collection, but some do require additional analyses.

Essential revisions:

1) Some conceptual issues. These can be addressed in the text.

a) The role of context as a modulator. The claim that the context modulates the PRF effect is not strong so it would be best to modify this claim. Firstly, the order of testing is not counterbalanced (mice tested in B before tested in A). Because of the order and because the difference in freezing between PRF and FRF during the baseline (i.e. context) in context B in Figure 1D does not seem to be very reliable (we don't see it in Figure 1B), it is hard to know what role the context really plays. That is, this masks potential differences in fear to A and B, which could account for the lack of PRF effect in Context A (through summation). What happens if the baseline for each mouse is subtracted from its individual CS freezing score? Is the PRF effect still lacking in context A? It is important to check this to see if the contextual effect is robust.

b) The DREADDs pathway study. The mPFC-BNST hM4Di data show that freezing in the PRF group is increased, reaching the levels of the FRF group. Could the lack of effect on FRF be due to ceiling levels? The trial by trial data may speak to this provided a decline in fear is seen across trials in at least one of the groups. Further, Figure 3H is suggestive of a difference between the FRF hM4Di and the mCherry groups. This matters because it may shed light on potential ceiling effect in the mPFC-BNST hM4Di behavioural data in the FRF groups. Further, an exciting difference would be between the mCherry groups. Was this tested?

2) Statistics.

a) Only overall ANOVAs are reported for the Fos+ analyses for each brain region, but the figures show asterisks suggestive of pairwise comparison tests. These should be reported in the text for CON, PRE and FRE for each brain area. This is also the case for the CS test in the mPFC-BNST pathway study.

b) Figure 3H is suggestive of a difference between the FRF hM4Di and the mCherry groups. Were the data analyzed together? An ANOVA seems appropriate here. Unsure why a Fisher's exact test was chosen and how it was conducted – for the 2x2 factorial or for individual cells of the table.

c) Direct comparisons between the PRF and FRF in the recording study are necessary.

d) Individual data points should be included in all the figures where possible.

3) Additional neural analyses.

a) It would be more helpful (beyond the rasters provided in Figure 3F) to see a summary heat plot of all recorded units, in all conditions, with activity aligned to cue onset/offset as well as freezing onset/offset. This will give the reader a much better idea of the patterns of activity observed in all neurons in all conditions. An example would be a heat plot like that in Figure 5C of Beyeler et al., 2018 Organization of valence-encoding and projection-defined neurons in the BLA, Cell Reports. If single-unit activity was suppressed by DREADD inhibition a heat plot would nicely illustrate this.

b) What group/condition was used for population firing in Figures 3G and 3I. Were these the PRF/mCherry neurons? Or was this all PRF neurons combined? (moot point for 3J if that is the case). This should not only be specified but the population should be plotted for all groups/conditions. This would allow the reader to see if cue responses to PRF and FRF were equivalent in the mCherry conditions – a result that is interesting no matter the outcome. This approach would provide less info for the data in 3J, because no PRF/hM4Di neurons were observed, but plotting the observed populations would still be helpful. The insets for 3G and 3I were not informative and should be removed as these neurons were selected because they were responsive.

c) Please provide waveform and firing rate properties for the neurons recorded. Did they differ between the groups? This is borne out of the between-subjects design of the recording experiments as different neuron types could have been sampled in each condition.

d) Full histology for electrode placements for all mice from which neurons were obtained should be provided. It would also be helpful to include schematics showing viral spread for all subjects (e.g., overlaid traces).

4) Materials and methods. Specific information about the number of mice and neurons recorded from in each group/condition need to be reported. Ranges are provided in the Figure 3 caption (n=17-25 units/group, from 3 mice per group/virus). But considering that the main statistics are based on the % of numbers, specific details need to be provided.

[Editors' note: further revisions were suggested prior to acceptance, as described below.]

Thank you for submitting your article "A prefrontal-bed nucleus of the stria terminalis circuit limits fear to uncertain threat" for consideration by *eLife*. Your article has been reviewed by two peer reviewers, including Mihaela D Iordanova as the Reviewing Editor and Reviewer #1, and the evaluation has been overseen by Kate Wassum as the Senior Editor.

The reviewers have discussed the reviews with one another and the Reviewing Editor has drafted this decision to help you prepare a revised submission.

Summary:

This is a revised version of the paper by Glover et al. We would like to reiterate the interest and excitement in the paper, which uncovers key brain areas and pathways that are involved in partial reinforcement in fear. While we appreciate that the authors have addressed the majority of the comments, the requests for additional analyses of the recording data were not well addressed well. The single-unit results cannot be meaningfully interpreted without additional analyses and visualizations as requested in the previous decision letter. Therefore, these concerns remain and we ask the authors to address them.

Essential revisions:

1) Please provide a summary heat plot of all recorded units, in all conditions, with activity aligned to cue onset/offset as well as freezing onset/offset. An example would be a heat plot like that in Figure 5C of Beyeler et al., 2018 Organization of valence-encoding and projection-defined neurons in the BLA, Cell Reports. This will give the reader a much better idea of the patterns of activity observed in all neurons in all conditions. If single-unit activity was suppressed by DREADD inhibition a heat plot would nicely illustrate this.

2) Population line graphs must be separately plotted for all groups/conditions when it is possible. This is necessary for the reader to see if cue responses to PRF and FRF were equivalent in the mCherry conditions – a result that is interesting no matter the outcome.

3) One other concern that remains is the possible ceiling levels in the DREADDs experiment. Ceiling in Figure 3—figure supplement 1 panel C and Figure 3—figure supplement 2 in panel C – Does the Figure 3—figure supplement 2C show the trial data for Figure 3—figure supplement 1C? This suggests ceiling across the board for all groups except PRF mCherry. This possibility needs to be reflected in the text as the possibility that the mPFC-BNST pathway could regulate fear in both cases remains.

4) Please report exact p values in all instances.

[Editors' note: further revisions were suggested prior to acceptance, as described below.]

Thank you for resubmitting your article "A prefrontal-bed nucleus of the stria terminalis circuit limits fear to uncertain threat" for consideration by *eLife*. Your article has been reviewed by two peer reviewers, including Mihaela D Iordanova as the Reviewing Editor and Reviewer #1, and the evaluation has been overseen by Kate Wassum as the Senior Editor.

The reviewers have discussed the reviews with one another and the Reviewing Editor has drafted this decision to help you prepare a revised submission.

Summary:

In a series of studies, Glover and colleagues examined the role of the prefrontal cortex and its input to the bed nucleus of stria terminalis pathway in fear to uncertain threats. The authors first demonstrate that partially reinforced cues support lower fear levels than fully reinforced cues. Comprehensive c-fos mapping is used to show that differential activation of the prefrontal cortex (prelimbic [PL] and infralimbic regions [IL]) and the bed nucleus of the stria terminalis (BNST) in threat uncertainty. DREADD inhibition of BNST-projecting mPFC neurons increased freezing to the partially reinforced cue. Electrophysiological examination show that mPFC neurons activated during PRF and FRF are reduced when the mPFC-BNST pathway is inactive. This is a second revision of the manuscript.

Essential revisions

We thank the authors for addressing all of the comments in the previous decision letter which pertained primarily to the analyses of the recording data. The reviewers have looked closely at the electrophysiology data and have noted that those data are rather preliminary in nature. Specifically, there are two main problems.

1) The data appeared underpowered: n=3 CS units in PRF/mCherry, n=6 units in PRF/hM4Di, n=1 units in FRF/mCherry, n=1 units in FRF/hM4Di.

2) The recording data only partially support the hypotheses of the paper. In some instances, the proportion of neurons do not support the hypotheses. For example, neither the heat plot nor the stats support the statement “Thus, IL cells were preferentially activated to the CS after PRF”. Further, the magnitude of the firing reveal a different pattern from the proportion. The former show

a) mPFC neurons show greatest CS onset firing to FRF

b) CS onset firing to FRF is diminished by hM4Di

c) mPFC neurons show equivalent firing increases to freezing cessation

d) Freezing cessation firing to FRF appears to be diminished by hM4Di

Other example of statements that are not supported by that data include, and therefore need to be revised:

"Thus, IL cells were preferentially activated to the CS after PRF and this activation was abolished by inhibition of the mPFC→BNST pathway."

"Thus, IL cells not only respond to presentation of the CS after PRF, but also to the ongoing behavior of the mouse and specifically during the transition from freezing to movement. As with CS responses, these neuronal correlates of behavior are absent when mPFC projections to the BNST are inhibited."

"it is tempting to conclude that this activity is a neuronal representation or even a causal driver of the lesser expression of PRF".

Ideally, the authors would have a full recording dataset. We recognize this may not feasible at this stage. So we give the authors two options: (1) To revise the language in the manuscript so that it can more accurately reflect the recording data, (2) To remove the behavioral electrophysiology data altogether.

---

## [Author Response]

Essential revisions:1) Some conceptual issues. These can be addressed in the text.a) The role of context as a modulator. The claim that the context modulates the PRF effect is not strong so it would be best to modify this claim. Firstly, the order of testing is not counterbalanced (mice tested in B before tested in A). Because of the order and because the difference in freezing between PRF and FRF during the baseline (i.e. context) in context B in Figure 1D does not seem to be very reliable (we don't see it in Figure 1B), it is hard to know what role the context really plays. That is, this masks potential differences in fear to A and B, which could account for the lack of PRF effect in Context A (through summation). What happens if the baseline for each mouse is subtracted from its individual CS freezing score? Is the PRF effect still lacking in context A? It is important to check this to see if the contextual effect is robust.

On reflection, we share the concerns regarding the open questions posed by these data and the experimental design used to obtain them – clearly follow up experiments are needed. Given their preliminary nature, we have removed them from the manuscript.

b) The DREADDs pathway study. The mPFC-BNST hM4Di data show that freezing in the PRF group is increased, reaching the levels of the FRF group. Could the lack of effect on FRF be due to ceiling levels? The trial by trial data may speak to this provided a decline in fear is seen across trials in at least one of the groups. Further, Figure 3H is suggestive of a difference between the FRF hM4Di and the mCherry groups. This matters because it may shed light on potential ceiling effect in the mPFC-BNST hM4Di behavioural data in the FRF groups. Further, an exciting difference would be between the mCherry groups. Was this tested?

We have looked at the trial by trial data for CS retrieval and there is no group difference (2x2 ANOVA P*>*.05) that would speak to the question of a potential ceiling effect in the FRF groups, as we now state as follows: “Examination of the trial by trial freezing during retrieval indicated no significant group differences, despite a trend for decreasing freezing across trials in the mCherry PRF group (ANOVA conditioning-type effect: P>.05; virus: P>.05; interaction: P>.05)…” The trend for an across-trial reduction in the FRF mCherry group that would be interesting to follow-up, for example with formal multi-trial extinction procedures.

2) Statistics.a) Only overall ANOVAs are reported for the Fos+ analyses for each brain region, but the figures show asterisks suggestive of pairwise comparison tests. These should be reported in the text for CON, PRE and FRE for each brain area. This is also the case for the CS test in the mPFC-BNST pathway study.

We now indicate pairwise differences (performed via Newman Keuls post hoc tests, as indicated in the Statistical analysis section), for example as follows: “Notably, however, CS-evoked freezing was lower in the PRF, relative to FRF, group (ANOVA group-effect: F(2,17)=53.02, P<.01; CS: F(1,17)=216.90, P<.01; interaction: F(2,17)=25.51, P<.01, followed by post hoc tests: CS vs PRF P<.05, CS vs FRF P<.05, PRF vs FRF P<.05).”

b) Figure 3H is suggestive of a difference between the FRF hM4Di and the mCherry groups. Were the data analyzed together? An ANOVA seems appropriate here. Unsure why a Fisher's exact test was chosen and how it was conducted – for the 2x2 factorial or for individual cells of the table.

We performed Fisher’s exact test to compare the percentage data in Figure 3H and 3J, rather than ANOVA, because these data are categorical and nonparametric – i.e., the percentage of cells with event-related activity in each of the 4 groups (meaning there is no variability). We do, however, now report an overall Fisher’s exact when all 4 groups are considered, as follows: “This revealed a higher proportion of CS-ON units in the mCherry than hM4Di groups for PRF mice (Fisher’s exact test: *P*<.05) but no differences for FRF mice (Fisher’s exact test: *P*>.05)…” and “When the percentage of these cell-types were compared across groups (Fisher’s exact test: *P*<.05), there was a higher percentage of Freeze-OFF units in the mCherry PRF group than in the hM4Di PRF group (Fisher’s exact test: *P*<.01), whereas there was no group difference in FRF mice (Fisher’s exact test: *P*>.05)…”

c) Direct comparisons between the PRF and FRF in the recording study are necessary.

We do, however, now report the Fisher’s exact comparing the PRF and FFR groups, as follows: “…and no difference between PRF and FRF groups, irrespective or virus group, despite a trend for (Fisher’s exact test in mCherry: *P*>.05; in hM4Di: *P*>.05) (Figure 3H)…” and “…and no difference between PRF and FRF groups in either the mCherry (Fisher’s exact test: *P*>.05) or hM4Di (Fisher’s exact test: *P*>.05) virus conditions (Figure 3J).”

d) Individual data points should be included in all the figures where possible.

As requested, individual data points are now included for the main figures, where possible.

3) Additional neural analyses.a) It would be more helpful (beyond the rasters provided in Figure 3F) to see a summary heat plot of all recorded units, in all conditions, with activity aligned to cue onset/offset as well as freezing onset/offset. This will give the reader a much better idea of the patterns of activity observed in all neurons in all conditions. An example would be a heat plot like that in Figure 5C of Beyeler et al., 2018 Organization of valence-encoding and projection-defined neurons in the BLA, Cell Reports. If single-unit activity was suppressed by DREADD inhibition a heat plot would nicely illustrate this.

While we agree that heat maps can provide a nice visual illustration of the temporal dynamics of unit activity, especially when driven by a temporally precise artificial stimulus as in the Beyeler et al. example, but we feel that showing these for all neurons is somewhat redundant to the example raster plot (Figure 3F) and peri-event histograms we already now show for all groups (Figure 3G,I), as requested (point #4 above).

b) What group/condition was used for population firing in Figures 3G and 3I. Were these the PRF/mCherry neurons? Or was this all PRF neurons combined? (moot point for 3J if that is the case). This should not only be specified but the population should be plotted for all groups/conditions. This would allow the reader to see if cue responses to PRF and FRF were equivalent in the mCherry conditions – a result that is interesting no matter the outcome. This approach would provide less info for the data in 3J, because no PRF/hM4Di neurons were observed, but plotting the observed populations would still be helpful. The insets for 3G and 3I were not informative and should be removed as these neurons were selected because they were responsive.

The data depicted in Figures 3G and 3I were all mice (PRF and FRF, mCherry and hM4Di) combined – the purpose to show the whole recorded population to first illustrate that IL neurons signal CS onset and freezing offset, and then subsequently show how this signaling varies between groups (Figures 3H and 3J). For this reason and because there are no differences between the percentage of phasic neurons in the mCherry groups (see point #5 above), we feel retain showing all groups combined and note this in the figure and figure legends. We maintain that even though they show event-responsive cells, the insets provided valuable information because they confirm that the event related activity is statistically robust (which does not necessarily follow from simply selecting cells that are event-responsive). Nonetheless, we have removed these and now report the values in the text, as follows: “Overall, CS-ON units showed a significant change in neuronal activity in response to the CS (baseline: 0.15±0.35, post-CS: 1.43±0.55, paired t-test: t(10)=6.51, P<.01) (Figure 3G).” And “These units, classified as Freeze-ON and Freeze-OFF, respectively, showed a significant change in baseline-normalized activity (Freeze-ON baseline: -0.81±0.47, post-event: -2.12±0.52, paired t-test: t(10)=4.60, P<.01, Freeze-OFF baseline: 0.73±0.43, post-event: 1.67±0.40, paired t-test: t(12)=8.54, P<.01) (Figure 3I).”

c) Please provide waveform and firing rate properties for the neurons recorded. Did they differ between the groups? This is borne out of the between-subjects design of the recording experiments as different neuron types could have been sampled in each condition.

The firing rates were given in the original manuscript, as follows “The average firing rate units did not differ between groups (FRF mCherry: 4.10±0.64, FRF hM4Di: 3.45±0.49, FRF mCherry: 2.67±0.66, FRF hM4Di: 1.67±0.34).”

While we agree that examining waveform in an effort to parse the molecular identity of the recorded cells would be of potential interest, but we do not feel confident that our data allow us to confidently explore this question given the small fraction of cells that would be putatively identifiable interneurons; i.e., it would be unlikely that we have sufficient numbers of interneurons recorded to make any strong inferences about their patterns of activity in this task. And though we recognize it is done in some studies, we did not perform waveforms analysis to categorize neurons as principal versus interneurons because we are not confident categorizing firing rate and waveform *per se* provides a reliable indicator of cortical cell type and would instead want to apply the current standard for categorization – optogenetic phototagging in an, e.g., parvalbumin-positive, interneuron Cre-driver line.

d) Full histology for electrode placements for all mice from which neurons were obtained should be provided. It would also be helpful to include schematics showing viral spread for all subjects (e.g., overlaid traces).

As requested, we now show schematics for electrode placements and viral spread.

4) Materials and methods. Specific information about the number of mice and neurons recorded from in each group/condition need to be reported. Ranges are provided in the Figure 3 caption (n=17-25 units/group, from 3 mice per group/virus). But considering that the main statistics are based on the % of numbers, specific details need to be provided.

The number of units and mice per group is now specified, as follows: “(n=17 units in PRF/mCherry, n=25 units in PRF/hM4Di, n=20 units in FRF/mCherry, n=17 units in FRF/hM4Di, from 3 mice per group/virus).”

[Editors' note: further revisions were suggested prior to acceptance, as described below.]

Essential revisions:1) Please provide a summary heat plot of all recorded units, in all conditions, with activity aligned to cue onset/offset as well as freezing onset/offset. An example would be a heat plot like that in Figure 5C of Beyeler et al., 2018 Organization of valence-encoding and projection-defined neurons in the BLA, Cell Reports. This will give the reader a much better idea of the patterns of activity observed in all neurons in all conditions. If single-unit activity was suppressed by DREADD inhibition a heat plot would nicely illustrate this.

We now provide a summary heat plot of all recorded units, in all conditions, with activity aligned to cue onset as well as freezing onset/offset in Figure 3—figure supplement 3. These provide a further visual demonstration of how DREADD inhibition suppresses CS and Freeze-OFF related activity, selectively in the PRF group – thank you for the useful suggestion.

2) Population line graphs must be separately plotted for all groups/conditions when it is possible. This is necessary for the reader to see if cue responses to PRF and FRF were equivalent in the mCherry conditions – a result that is interesting no matter the outcome.

As requested, line graphs are now separately plotted for all groups/conditions in Figure 3 and Figure 3—figure supplement 7. We present these along with the original all-group averages and discuss the graphs as follows: “Peak responses occurred withing 200-300 seconds of CS onset and were highest in the mCherry FRF group (Figure 3H). However, when the percentage of CS-ON units was calculated and compared across the conditioning and virus groups (Fisher’s exact test: *P*<.05), this revealed a higher proportion of CS-ON units in the mCherry than hM4Di groups for PRF mice (Fisher’s exact test: *P*=.0122), but no differences between virus groups in the FRF mice (Fisher’s exact test: *P*=.6090), and no difference between PRF and FRF groups, irrespective or virus group (Fisher’s exact test in mCherry: *P*=.2510; in hM4Di: *P*=1.000) (Figure 3I),” and “Freeze-ON units displayed a decreased firing rate at freezing onset, which was most evident in both of the mCherry groups, while Freeze-OFF units increased firing rate at the cessation of freezing in both groups. (Figure 3K, Figure 3—figure supplement 3).”

3) One other concern that remains is the possible ceiling levels in the DREADDs experiment. Ceiling in Figure 3—figure supplement 1 panel C and Figure 3—figure supplement 2 in panel C – Does Figure 3—figure supplement 2C show the trial data for Figure 3—figure supplement 1C? This suggests ceiling across the board for all groups except PRF mCherry. This possibility needs to be reflected in the text as the possibility that the mPFC-BNST pathway could regulate fear in both cases remains.

That is correct, Figure 3—figure supplement 2C shows the trial data for Figure 3—figure supplement 1C. In the text, we now refer to the figure, as follows: “Examination of the trial-by-trial freezing during retrieval indicated no significant trial-related differences in freezing, despite a trend for decreasing freezing across trials in the mCherry PRF group (ANOVA trial-effect: F(5,145)=1.83, *P*=.1098; group-effect: F(3,29)=14.15, *P*<.0001; trial x group interaction: F(15,145)=1.04, *P*=.4213) (Figure 3—figure supplement 1).” We also now discuss the possibility of a ceiling effect, as follows: “These data show that inhibition of mPFC→BNST neurons increases freezing to a PRF CS. This finding suggests engagement of these mPFC→BNST neurons limits the expression of PRF, though it remains possible that inhibition of these neurons also produces an increase in PRF expression which may have been masked due to high (“ceiling”) levels of freezing.”

4) Please report exact p values in all instances.

All p values are now reported, up to a threshold of *P*=.0001, as stated under “Statistical analysis,” as follows: “The threshold for statistical significance was set at *P*<0.05; significance values are shown up to *P*<.0001.”

[Editors' note: further revisions were suggested prior to acceptance, as described below.]

Essential revisionsWe thank the authors for addressing all of the comments in the previous decision letter which pertained primarily to the analyses of the recording data. The reviewers have looked closely at the electrophysiology data and have noted that those data are rather preliminary in nature. Specifically, there are two main problems.1) The data appeared underpowered: n=3 CS units in PRF/mCherry, n=6 units in PRF/hM4Di, n=1 units in FRF/mCherry, n=1 units in FRF/hM4Di.2) The recording data only partially support the hypotheses of the paper. In some instances, the proportion of neurons do not support the hypotheses. For example, neither the heat plot nor the stats support the statement “Thus, IL cells were preferentially activated to the CS after PRF”. Further, the magnitude of the firing reveal a different pattern from the proportion. The former showa) mPFC neurons show greatest CS onset firing to FRFb) CS onset firing to FRF is diminished by hM4Dic) mPFC neurons show equivalent firing increases to freezing cessationd) Freezing cessation firing to FRF appears to be diminished by hM4DiOther example of statements that are not supported by that data include, and therefore need to be revised:"Thus, IL cells were preferentially activated to the CS after PRF and this activation was abolished by inhibition of the mPFC→BNST pathway.""Thus, IL cells not only respond to presentation of the CS after PRF, but also to the ongoing behavior of the mouse and specifically during the transition from freezing to movement. As with CS responses, these neuronal correlates of behavior are absent when mPFC projections to the BNST are inhibited.""it is tempting to conclude that this activity is a neuronal representation or even a causal driver of the lesser expression of PRF".Ideally, the authors would have a full recording dataset. We recognize this may not feasible at this stage. So we give the authors two options: (1) To revise the language in the manuscript so that it can more accurately reflect the recording data, (2) To remove the behavioral electrophysiology data altogether.

In view of the reviewers’ concerns about the number of recorded units showing event-related activity, we have moved in vivo unit recording data to the supplemental materials and revised the language in the manuscript by:

Removing the following statements: “Thus, IL cells were preferentially activated to the CS after PRF and this activation was abolished by inhibition of the mPFC→BNST pathway,” “Thus, IL cells not only respond to presentation of the CS after PRF, but also to the ongoing behavior of the mouse and specifically during the transition from freezing to movement. As with CS responses, these neuronal correlates of behavior are absent when mPFC projections to the BNST are inhibited” and “it is tempting to conclude that this activity is a neuronal representation or even a causal driver of the lesser expression of PRF.”

Changing the title of the corresponding Results section from “IL cells preferentially signal CS-onset and freezing cessation to a PRF CS” to “IL cells signal CS-onset and freezing cessation.”

The relevant statement in the Abstract to “Multiplexing chemogenetics with in vivo neuronal recordings showed elevated infralimbic cortex (IL) neuronal activity during CS-onset and freezing-cessation; these neural correlates were abolished by chemogenetic mPFC→BNST inhibition.”

And made other changes in the Discussion to avoid making claims about the specify of the unit correlates to PRF; for example “Intriguingly, we also found a subset of IL neurons that displayed phasic activity during the cessation, but not onset, of freezing during [add “fear”] retrieval, echoing recordings in rat…”